# Distinct Impacts of Two Types of Developing El Niño–Southern Oscillations on Tibetan Plateau Summer Precipitation

**Minghong Liu, Hong-Li Ren * , Run Wang, Jieru Ma and Xin Mao**

State Key Laboratory of Severe Weather, and Institute of Tibetan Plateau Meteorology,
Chinese Academy of Meteorological Sciences, Beijing 100081, China
* Correspondence: renhl@cma.gov.cn

**Abstract:** El Niño–Southern Oscillation (ENSO) has remarkable impacts on Tibetan Plateau (TP) summer precipitation. However, recently identified ENSO spatial diversity brings complexity to these impacts. This study investigates the distinct impacts of the Eastern Pacific (EP) and Central Pacific (CP) ENSOs on TP summer precipitation based on numerous precipitation data and satellite-observed and reanalyzed circulation data. The results show that the EP El Niño and the CP La Niña have opposite effects on summer precipitation in the southwestern TP, with significant decreases and increases, respectively, indicating a trans-type inversion. In contrast, the CP El Niño causes significant decreases in summer precipitation in the central-eastern TP. No significant anomaly occurs during the EP La Niña. Moisture budget and circulation analyses suggest that these distinct precipitation characteristics can be attributed to different atmospheric teleconnections, which provide varying vertical motion and moisture conditions. The EP El Niño triggers an atmospheric response similar to the Indian Summer Monsoon–East Asian Summer Monsoon teleconnection, and the CP La Niña generates a teleconnection in the opposite phase. In contrast, the CP El Niño mainly causes a teleconnection resembling the East Asian–Pacific pattern. This study may deepen our understanding of ENSO impacts on TP summer precipitation and have implications for regional climate predictions.

**Keywords:** Tibetan Plateau precipitation; El Niño–Southern Oscillation; ENSO diversity; ENSO teleconnection

## 1. Introduction

The Tibetan Plateau (TP), known as the "third pole", is the largest and highest plateau in the world, with an average elevation of approximately 4000 m and the largest non-polar ice reservoir [1,2]. Given its unique elevation and geographic position, TP plays a critical role in shaping and interacting with the global climate via dynamical blocking and thermal effects on large-scale atmospheric circulations [3–6]. The TP is also well-known as the "Asian water tower" because of its substantial role in the hydroclimate, which supplies water resources to the streamflow and runoff of several major rivers in Asia, including the Yangtze River, the Yellow River, and the Mekong River. Thus, the TP has a notable impact on the balance of downstream ecosystems and anthropogenic activities and livelihoods [7,8].

Precipitation is one of the most important factors modulating the TP hydroclimate, especially during its rainy season [9,10]. The variation in TP precipitation significantly impacts large-scale atmospheric circulations and climate variability in downstream East Asia and even globally through enormous latent heat releases and associated heating feedback [4,5,11]. Therefore, it is of great importance to investigate the interannual variability of TP precipitation and understand the underlying mechanisms.

The rainy season of the TP generally starts in May at the onset of the boreal (omitted in the following) summer monsoon and lasts until September, accounting for around 80 per-cent of the total annual precipitation [9,12]. Despite the limited data available on TP precipitation, many efforts have been devoted to studying the external factors that

contribute to the interannual variation of TP summer precipitation, such as the summer North Atlantic Oscillation [13,14] and the Indian summer monsoon (ISM) [15–17]. Sea surface temperature (SST) anomalies in the Atlantic Ocean and Indian Ocean also modify TP summer precipitation [9,18–20]. The El Niño–Southern Oscillation (ENSO), the dominant mode of interannual variability in the tropical Pacific Ocean, has also been revealed to impact TP summer precipitation remotely [21–23]. However, the impact of ENSO on TP summer precipitation is typically indirect and mediated by various factors. Previous studies mainly focused on TP precipitation anomalies during the decaying phase of ENSO. El Niño-associated basin-wide warming of the Indian Ocean SST may significantly delay the outbreak of the South Asian summer monsoon, thus reducing precipitation over the southeastern TP in the early rainy season [9,18,24,25]. The Indian Ocean basin-wide SST anomaly can also impact TP precipitation by adjusting the meridional position of the South Asian High (SAH) [26,27]. Moreover, the SST anomaly over the tropical northern Indian Ocean and western Pacific Ocean can persist during summer through the local air–sea interaction. This persistent SST anomaly has been revealed to play an important role in maintaining and enhancing the anomalous anticyclonic circulation over the western North Pacific during the decaying summer of El Niño, known as the Indo-western Pacific Ocean capacitor mechanism [28,29]. This enhanced anticyclonic circulation leads to increased water vapor transported to the TP, thus increasing summer TP precipitation [30,31].

Furthermore, some studies link the TP summer precipitation anomaly to the Pacific SST variation during the developing summer of ENSO [21,22]. ENSO-induced anomalous convective heating directly modifies the upper-tropospheric thermal status and influences the subtropical westerly winds covering the TP, thereby impacting its precipitation [32,33]. The developing ENSO also influences TP precipitation through its teleconnection impacts in the Indian Ocean. Many studies have suggested that ENSO can influence the intensity of ISM precipitation by altering the Walker circulation, and the suppressed ISM precipitation stimulates an upper-tropospheric cyclonic circulation anomaly to the west of the TP as part of the ISM-East Asian Summer Monsoon (EASM) teleconnection [34–36], which significantly reduces precipitation in the southwestern TP (SWTP) [15,37]. Meanwhile, the altered Walker circulation can also strengthen the India–Burma monsoon trough through regional air–sea interaction over the Indo-western Pacific Ocean [38,39], modifying the moisture transported to the TP and thereby affecting TP precipitation. In addition, ENSO-associated variations in TP local variables, such as the snow cover and water reserve, also modulate TP precipitation through coupled feedback [40,41].

Besides the various direct and indirect mechanisms of climate impacts, ENSO itself has characteristics of spatial diversity, which adds complexity to the topic of ENSO impacts [42–45]. In recent decades, a new type of El Niño, namely, the Central–Pacific (CP) El Niño, has frequently occurred. Compared with the canonical Eastern–Pacific (EP) El Niño, the SST anomaly center of the CP El Niño is westward shifted. Many studies have emphasized the substantial influence of ENSO diversity on its climate impacts [46–50]. For example, evident differences have been found between the impacts of the two types of El Niños on precipitation and air temperature in China [51–53]. Similarly, La Niña can also be classified into two types that exhibit distinct local and teleconnection responses based on their SST anomaly patterns [54], although whether these two types of La Niña are intrinsically distinguished remains controversial [55].

Some intermediate factors and processes linking ENSO and TP precipitation, such as the Indian Ocean air–sea interaction, exhibit distinct responses to different types of EN-SOs [56,57]. However, the resulting variation of the TP precipitation response to ENSO has received less attention. A quasi-three-year cycle of interannual variability in precipitation (less–more–more) during the Tibetan rainy season was once proposed, which corresponds to an evolution consisting of the development of an EP El Niño, a shift from a CP El Niño to an EP La Niña, and the decaying of a CP La Niña [22]. However, realistic ENSO events do not always follow such an evolution. The precipitation response to different types of ENSOs remains unclear. This study aims to investigate TP summer precipitation variation due to

ENSO spatial diversity and the underlying mechanisms from the perspective of historical ENSO events. For practice, this study focuses on the developing summer of ENSO.

The remainder of the paper is organized as follows. Section 2 introduces the data and method utilized in this study. Section 3.1 presents the characteristics of the TP summer precipitation anomaly during different types of ENSOs, and the causes for these distinct characteristics are explored in Section 3.2 by examining large-scale circulation and moisture anomalies. Section 4 provides a brief discussion. The main conclusions are finally listed in Section 5.

## 2. Data and Methods

Four high-resolution gridded precipitation datasets are employed in this study, including Climate Research Unit Time-Series (CRU) version 4.05 [58], Global Precipitation Climatology Centre (GPCC) full data reanalysis version 2020 [59], CN05.1 precipitation [60], and Asian Precipitation-Highly–Resolved Observational Data Integration Towards Evaluation (APHRODITE) gridded precipitation versions 1101 and 1101EX-R1 [61]. These datasets are mainly based on dense gauge data networks, and some have undergone satellite-based quality control. We also perform an ensemble mean of these precipitation datasets using equal weights to remove inter-dataset bias. Observational precipitation data from 2400 stations of the China Meteorological Data Sharing Service System are used as a reference for the grid datasets.

The data for the outgoing longwave radiation (OLR), which is commonly used as a remote sensing proxy for convective activity, are obtained from the Interpolated Outgoing Longwave Radiation dataset provided by the National Oceanic and Atmospheric Administration (NOAA) Physical Sciences Laboratory [62]. Other atmospheric variables are from the European Centre for Medium-Range Weather Forecasts reanalysis generation 5 (ERA5) [63], and the SST data is obtained from the HadISST supplied by the Met Office Hadley Center [64]. All data utilized in this study are re-interpolated into a horizontal resolution of $1° \times 1°$ in a longitude/latitude grid. The analyzing period ranges from January 1961 to December 2020 (to December 2015 for the APHRODITE precipitation data), ensuring sufficient coverage of historical ENSO events and relatively reliable data. Note that the OLR data from NOAA starts in 1975, and data from ERA5 complements the earlier period. A linear trend was removed before analysis. Anomalies are obtained by removing a climatological mean derived from the whole analyzing period and conducting a 3-month running average to filter the noise. A composite analysis is utilized to investigate the common features of specific events. The statistical significance of our results is determined using the two-tailed Student's *t*-test. The "summer" in this study is defined as June–July–August–September (JJAS), the significant rainy season of the TP. The qualitative conclusions remain the same if using June–July–August or May–September.

ENSO events are identified when the 3-month running mean of the Niño3.4 index (SST anomalies averaged in the region of 5°S–5°N, 170°–120°W) exceeds a threshold of 0.5 °C for a minimum of five consecutive months [53]. Then we classify the identified ENSO events based on the spatial distribution of SST anomalies during JJAS preceding the ENSO peak, from which both spatial types of El Niño or La Niña can be effectively distinguished [54]. Specifically, events with the largest SST anomaly center located east (or west) of 150°W are classified into the EP (or CP)-type developing ENSO. In the analyzing period, a total of ten EP-type developing El Niños (1963, 1965, 1969, 1972, 1976, 1982, 1987, 1997, 2009, and 2015) and seven CP-type developing El Niños (1968, 1977, 1991, 1994, 2002, 2004, and 2006) are selected. Meanwhile, there are ten EP-type developing La Niñas (1964, 1970, 1973, 1984, 1988, 1995, 2007, 2010, 2017, and 2020) and five CP-type developing La Niñas (1975, 1999, 2000, 2008, 2011). The El Niño event in 2018 did not develop until autumn, and the La Niña event in 1983 developed accompanying a strong warming SST anomaly in the far-eastern tropical Pacific Ocean; these two events are excluded for clearly displaying the TP summer precipitation response to the developing ENSO. Moreover, to quantitatively measure the spatial characteristics of ENSO, the cold-tongue index (CTI) and warm-pool index (WPI)

are calculated [45]. CTI is defined as $N_3 - \alpha \times N_4$, and WPI is $N_4 - \alpha \times N_3$, where $N_3$ is the mean SST anomalies over the Niño3 region (5°S–5°N, 150°–90°W) and $N_4$ is over the Niño4 region (5°S–5°N, 160°E–150°W). Parameter $\alpha$ is 0.4 if $N_3 \times N_4 > 0$; otherwise, $\alpha$ is set to 0.

An atmospheric moisture budget analysis [37] is conducted to understand the processes responsible for regional precipitation anomalies based on the following equation:

$$P' = E' - < u\frac{\partial q}{\partial x} >' - < v\frac{\partial q}{\partial y} >' - < \omega\frac{\partial q}{\partial p} >' + \xi \tag{1}$$

where the prime denotes the monthly anomaly, and the bracket denotes the vertical integration from the surface to the top of the atmosphere. $P$, $E$ and $\xi$ are the precipitation, evaporation and residual terms, and the others are advection terms. $u$, $v$, $\omega$, and $q$, separately represent the zonal and meridional winds, the vertical pressure velocity, and the specific humidity. The term $- < \omega\frac{\partial q}{\partial p} >'$ can be further decomposed into two linear advection terms ($- < \omega'\frac{\partial \bar{q}}{\partial p} >$ and $- < \bar{\omega}\frac{\partial q'}{\partial p} >$) corresponding to the anomalous vertical motion and local moisture variation separately, and a nonlinear term consisting of nonlinear advection and a sub-monthly transient ($- < \omega'\frac{\partial q'}{\partial p} > + \text{sub}$). The other two advection terms can be decomposed similarly.

## 3. Results

### 3.1. TP Summer Precipitation Anomalies during Different Types of ENSOs

We first show composites of TP precipitation anomalies during the developing summer of different types of ENSOs (Figure 1). The figures presented in the main text are all based on the ensemble mean of four gridded precipitation datasets, and the main conclusions of this study remain consistent across all datasets (Figure S1). If not distinguishing spatial types of ENSOs, TP summer precipitation generally decreases relative to the climatological mean during El Niño events, especially in the southwest and east (Figure 1a). This result is consistent with previous studies suggesting that SWTP summer precipitation is negatively correlated with simultaneous SST warming anomalies in the tropical eastern Pacific, e.g., [22]. Figure 1b,c further show the precipitation anomaly pattern corresponding to the two spatial types of El Niños, and it is clear that the two anomaly centers in Figure 1a are attributed to different types of El Niño. Specifically, the negative summer precipitation anomaly in the SWTP mainly occurs during the EP El Niños (Figure 1b), while the extensive precipitation decrease in the central-eastern TP (CETP) primarily exhibits during the CP El Niño (Figure 1c). The latter is rarely noted in previous studies.

The situation differs for TP precipitation anomalies during the developing summer of La Niña. Significant positive anomalies only occur in the SWTP in the composite for all La Niña events, while nothing is observed for the CETP (Figure 1d). After further classifying their spatial types, no significant anomalies can be observed during the EP La Niña, while CP events accompany statistically enhanced SWTP summer precipitation. Combining the results during El Niños, we can conclude that both EP and CP types of ENSO exhibit asymmetric impacts on TP summer precipitation between El Niño and La Niña phases. Moreover, the TP summer precipitation anomaly pattern during the CP La Niña appears to be opposite to that observed during the EP El Niño rather than the CP El Niño, suggesting a trans-type inversion. The anomaly in the CETP is unique to the CP El Niño.

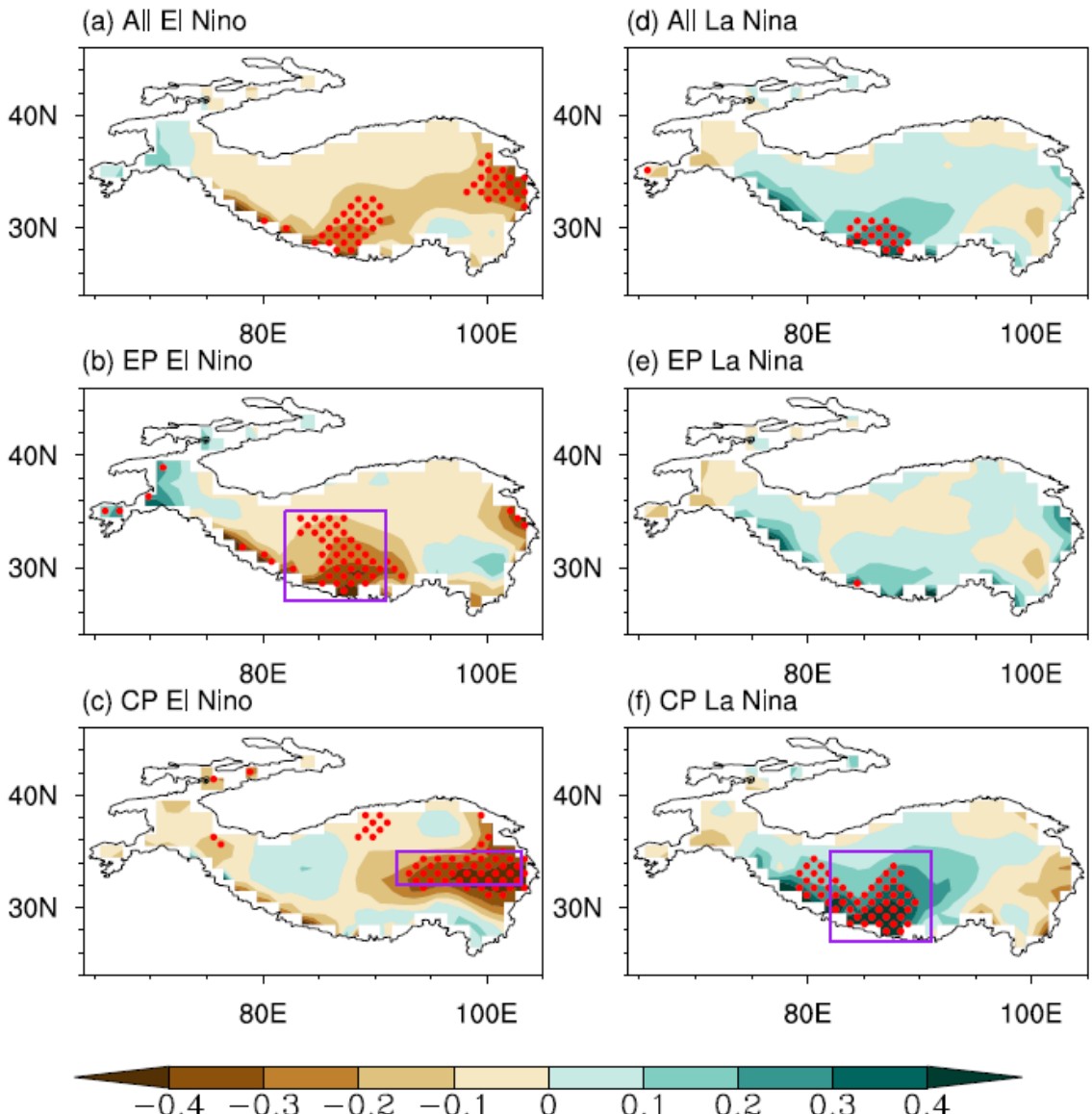

**Figure 1.** Composites of JJAS precipitation (unit: mm·day$^{-1}$) over the TP for different types of ENSO events. The red dotted areas are statistically significant at the 90% confidence level.

The asymmetric impacts due to ENSO diversity can be directly reflected in the spatial distribution of correlation between TP summer precipitation and the simultaneous Niño3.4 index. The correlation is high in the SWTP, while it is very low in the CETP (Figure 2a). We further define two regional precipitation indices to examine the relationship between TP summer precipitation and different types of ENSOs event by event. One is an SWTP precipitation index calculated as the area mean of the summer precipitation anomaly over the region located at 27°–35°N, 82°–91°E, and another is a CETP precipitation index similarly defined as the mean anomaly over the region located at 32°–35°N, 92°–103°E. Purple boxes mark these two regions in Figure 1. Even though SWTP summer precipitation strongly depends on ENSO intensity with a linear coefficient of around −0.66 within all ENSO events (Figure 2b), the role of ENSO diversity can still be observed. It is mainly EP ENSO events that contribute to the linear correlation, while the CP ENSO's impacts are nonlinear. Specifically, SWTP summer precipitation decreases during all EP El Niño events except that in 1963, but no regularity presents during CP El Niño events. For La Niñas, SWTP summer precipitation increases during only some strong EP events, while it increases during all CP events even with relatively weak ENSO intensity.

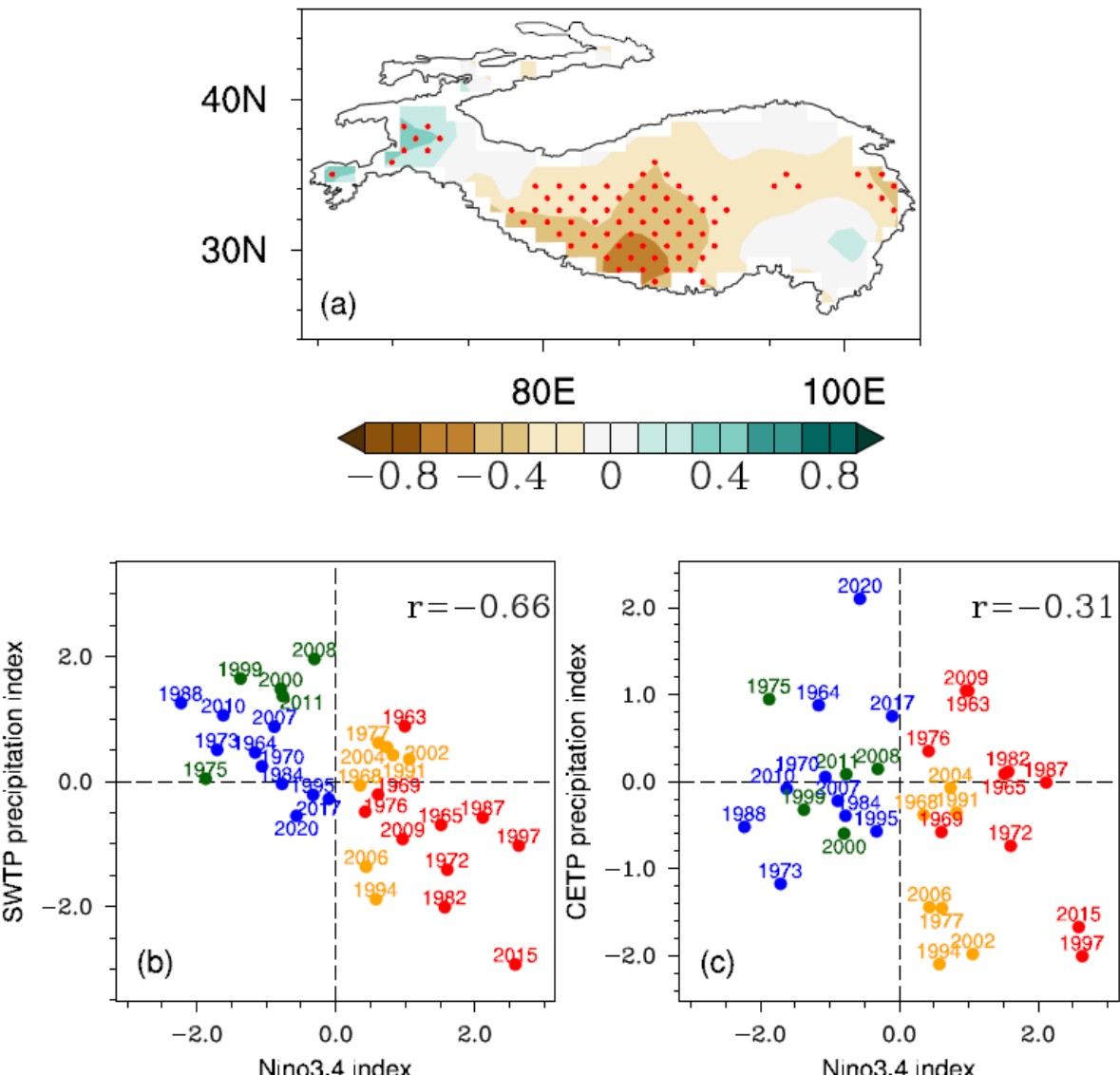

**Figure 2.** (**a**) Correlation distribution between TP summer precipitation anomalies and the Niño3.4 index (unit: 1). Red dots indicate correlation coefficients statistically significant at the 90% confidence level. The scatterplots show the normalized (**b**) SWTP precipitation index (unit: 1) and (**c**) CETP precipitation index (unit: 1) as a function of the normalized Niño3.4 index (unit: 1) in JJAS for all ENSO events. The red, orange, blue, and green solid circles separately indicate EP El Niño, CP El Niño, EP La Niña, and CP La Niña events. The colored text notes the developing year of ENSO events. The text in the bottom right presents the linear correlation coefficient between the TP precipitation index and the Niño3.4 index for all ENSO events.

In contrast, CETP summer precipitation is less connected with ENSO intensity, with a linear correlation coefficient of just −0.31 (Figure 2c). It only regularly decreases during CP El Niño events but randomly varies during other types of developing ENSOs. The above analyses highlight the role of ENSO spatial diversity in impacting TP summer precipitation, which makes these ENSO impacts asymmetric and trans-type inverse.

### 3.2. Circulation and Moisture Anomalies Associated with Different Types of ENSOs

How do these distinct TP summer precipitation anomalies form during different types of developing ENSOs? On an interannual timescale, vertical motion and large-scale moisture transport are crucial factors controlling TP summer precipitation variation [37].

Figure 3 illustrates anomalous summer upper-tropospheric circulation during different types of ENSOs. During the EP El Niño, a pair of cyclonic circulation anomalies characterize the subtropical upper troposphere. These anomalies are located to the west of the TP and over northern China, respectively, and are associated with evident negative geopotential anomalies (Figure 3a). These cyclonic circulation anomalies are equivalent-barotropic, with cyclonic circulation anomalies occurring at the low level (Figure 4a). These two equivalent-barotropic cyclonic circulation anomalies relate the ISM to the EASM, referred to as the ISM-EASM teleconnection [34,35]. Meanwhile, positive geopotential anomalies exist over the northern Indian Ocean and western North Pacific regions. These low-latitude anomalies are the Kelvin and Rossby wave responses directly induced by the enhanced convection in the tropical Pacific associated with El Niño [32]. In this case, significant westerly wind anomalies occur to the south of the two cyclonic circulation anomalies, crossing the SWTP (Figure 3a). This anomaly pattern suggests a southward-shifted SAH and a subtropical westerly jet that cause descending motion anomalies over the SWTP, thus decreasing rainfall [24–27].

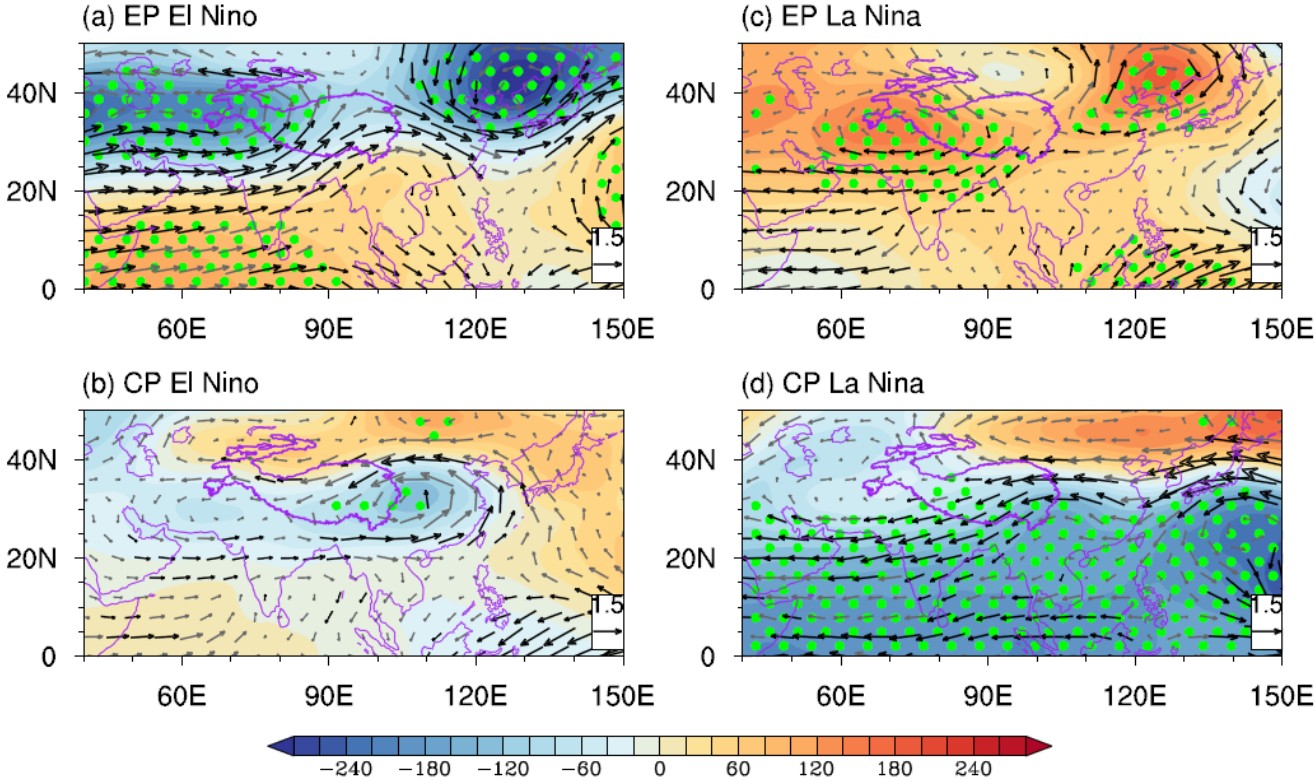

**Figure 3.** Composites of JJAS-mean 200-hPa geopotential (shading, unit: $m^2 \cdot s^{-2}$) and wind anomalies (vectors, unit: $m \cdot s^{-1}$) for different types of ENSO events. Green dots indicate geopotential anomalies statistically significant at the 90% confidence level. Dark-colored vectors denote the wind anomalies whose zonal or meridional components are significant at the 90% confidence level.

However, the situation changes during the developing summer of CP El Niño. Compared with those of EP El Niño, upper-tropospheric positive geopotential anomalies over the northern Indian Ocean and anomalous cyclonic circulation to the west of the TP disappear. Instead, the CP-El Niño-associated circulation pattern manifests as a significant meridional equivalent-barotropic dipole over mid-latitude East Asia, which features a significant cyclonic circulation anomaly over eastern China and an anticyclonic circulation anomaly north of it (Figures 3b and 4b). This circulation pattern resembles the so-called East Asia–Pacific (EAP) pattern (also known as the Pacific–Japan pattern) [65–67]. This result is reasonable, as previous studies have implied that the EAP pattern is more clearly shown in the developing summer of CP ENSO rather than EP events [65,68]. This EAP-like pattern's

upper-tropospheric cyclonic circulation anomaly covers the CETP with significant easterly wind anomalies, suggesting an abnormal descending motion. Therefore, the significant region of the precipitation anomaly switches to the CETP during the CP El Niño (Figure 1c).

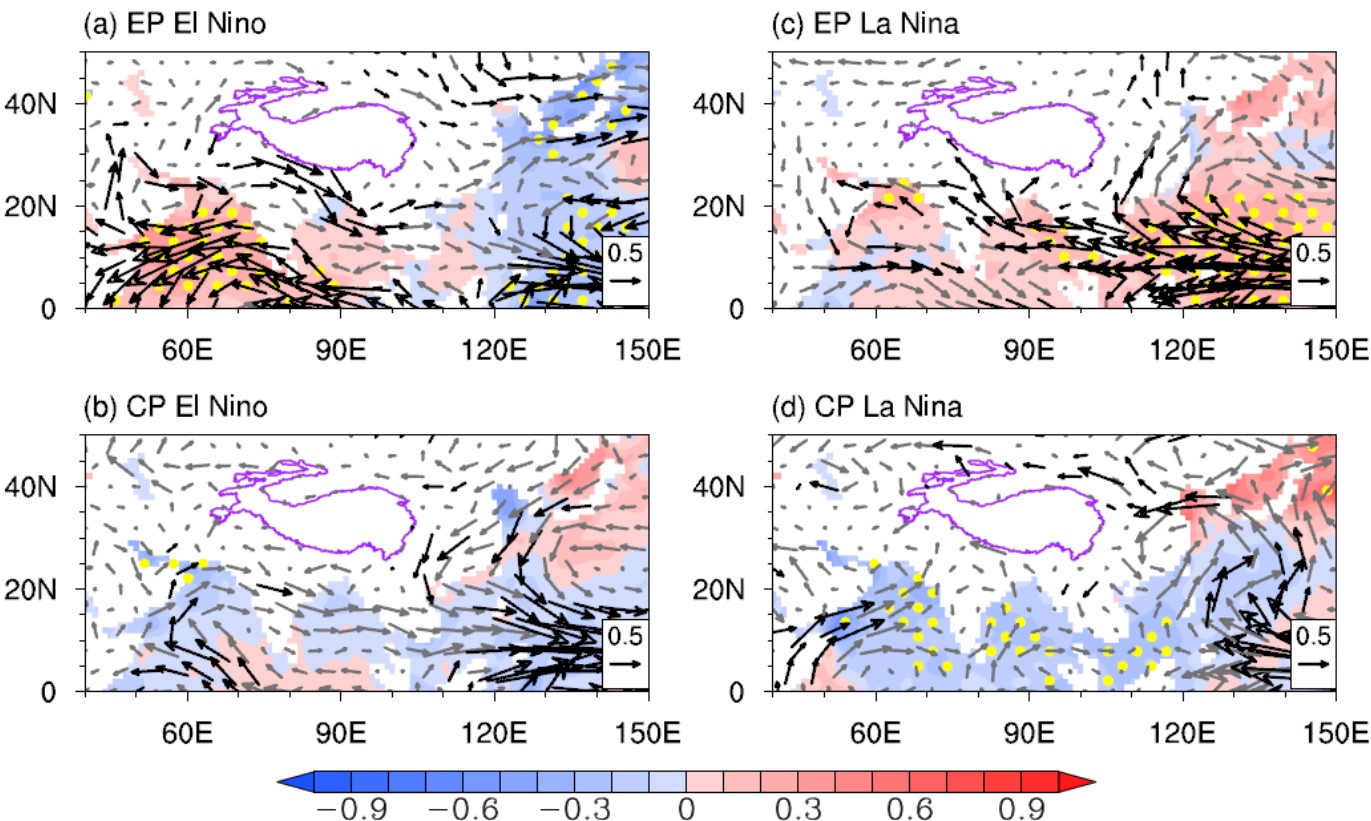

**Figure 4.** Composites of JJAS-mean SST (shading, unit: °C) and 850-hPa wind (vectors, unit: $10 \text{ m·s}^{-1}$) anomalies for different types of ENSO events. Yellow dots indicate SST anomalies statistically significant at the 90% confidence level. Dark-colored vectors denote the wind anomalies whose zonal or meridional components are significant at the 90% confidence level.

During the developing summer of EP La Niña events, a pair of significant equivalent-barotropic anticyclonic circulation anomalies locate west of the TP and over northern China, respectively, and the western one covers the SWTP (Figures 3c and 4c). Therefore, the SWTP might be controlled by anomalous ascending motion. This upper-tropospheric circulation anomaly can also be seen as an ISM-EASM teleconnection pattern featuring anticyclonic circulation anomalies. However, it should be noted that there are few negative geopotential anomalies in the upper troposphere over the subtropical Indo-western Pacific Ocean region, suggesting a relatively weak response of the tropical upper troposphere to the EP La Niña. In this case, less easterly wind anomalies occur over subtropical Eurasia, mainly northerly wind anomalies dominating the SWTP. Several previous studies have suggested that the variation in the subtropical westerly jet related to the zonal wind anomaly is a key factor that affects anomalous vertical motion and precipitation over the SWTP [25,26,37]. Thus, the circulation anomaly associated with EP La Niña events has little influence on SWTP precipitation.

During the CP La Niña, the upper-tropospheric circulation anomaly pattern features intense negative geopotential anomalies over the low-latitude Indo-western Pacific Ocean region and a pair of equivalent-barotropic anticyclonic circulation anomalies located to the west of the TP and over northern China, although the one west of the TP is less significant (Figures 3d and 4d). This pattern is essentially an ISM-EASM teleconnection pattern of the opposite phase to its counterpart during the EP El Niño. In this case, strong easterly wind

anomalies dominate subtropical Eurasia, including the SWTP. This circulation anomaly pattern may suggest a northward-shifted SAH [26] with an abnormal ascending motion over the SWTP.

We also examine anomalous vertically-integrated vapor flux during the developing summer of different types of ENSOs. As shown in Figure 5a, an intense anomalous southeastward vapor flux occurs south of the TP during EP El Niño events. This anomalous flux suppresses the ISM precipitation and hinders the import of moisture to the TP from its southwestern boundary. Meanwhile, anomalous eastward vapor flux dominates moisture transport over the TP under strong upper-tropospheric westerly wind anomalies, also contributing to a significant moisture deficiency over the SWTP. Therefore, besides the effect of anomalous descending motion, the anomalous moisture transport during EP El Niño events also benefits the decrease in SWTP precipitation. In addition, the suppressed ISM results in an extensive negative convection anomaly over the Indian subcontinent (Figure 6a), which has been revealed as a key factor in triggering the ISM-EASM teleconnection [15,35].

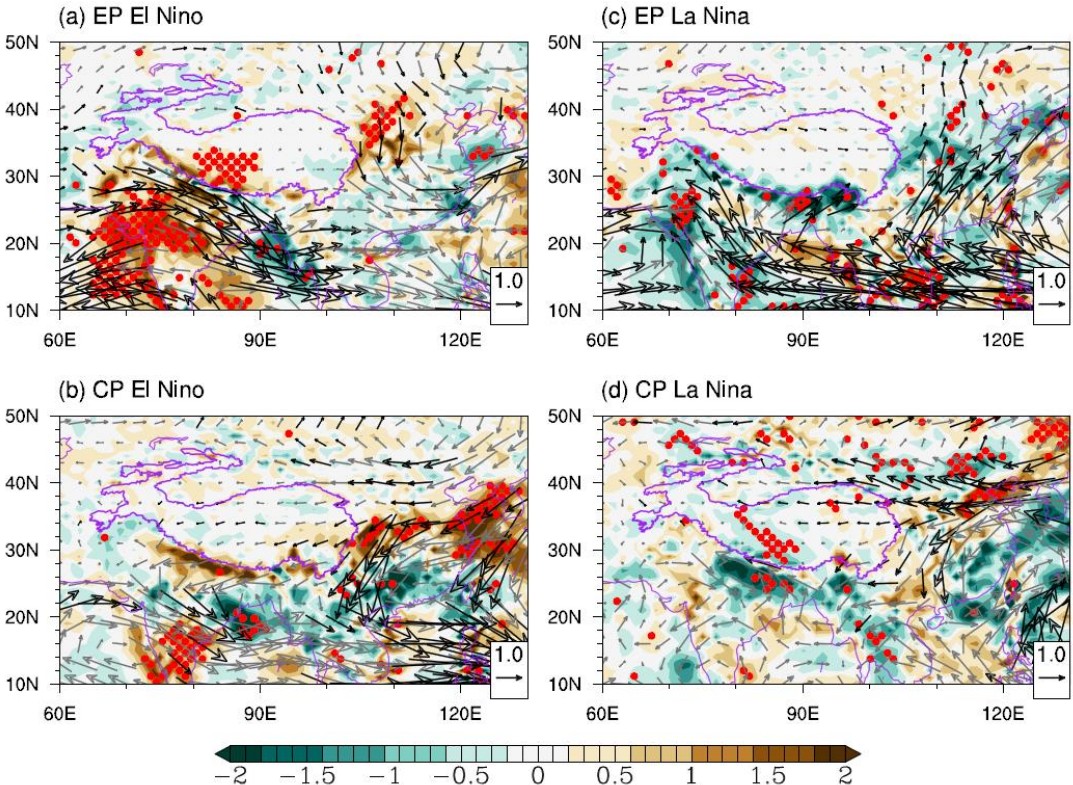

**Figure 5.** Composites of JJAS-mean vertically integrated vapor flux (vectors, unit: $10\ \mathrm{kg \cdot m^{-1} \cdot s^{-1}}$) and its divergence (shading, unit: $10^{-5}\ \mathrm{kg \cdot m^{-2} \cdot s^{-1}}$) anomalies for different types of ENSO events. Red dots indicate divergence anomalies statistically significant at the 90% confidence level. Dark-colored vectors denote the flux anomalies whose zonal or meridional components are significant at the 90% confidence level.

During the CP El Niño, in contrast, the anomalous moisture transport from the Indian Ocean and the suppressed convection on the Indian subcontinent are not significant (Figures 5b and 6b). The vapor flux anomaly imported to the TP mainly comes from eastern China and the North Pacific, consistent with the upper-tropospheric easterly wind anomalies over the TP (Figure 3b). However, the moisture convergence anomaly over the CETP is negligible, partly because the regional descending motion anomaly may suppress the moisture convergence there. This result suggests that unlike during the EP El Niño, anomalous moisture transport may play a minor role in causing negative CETP precipitation anomalies during the CP El Niño.

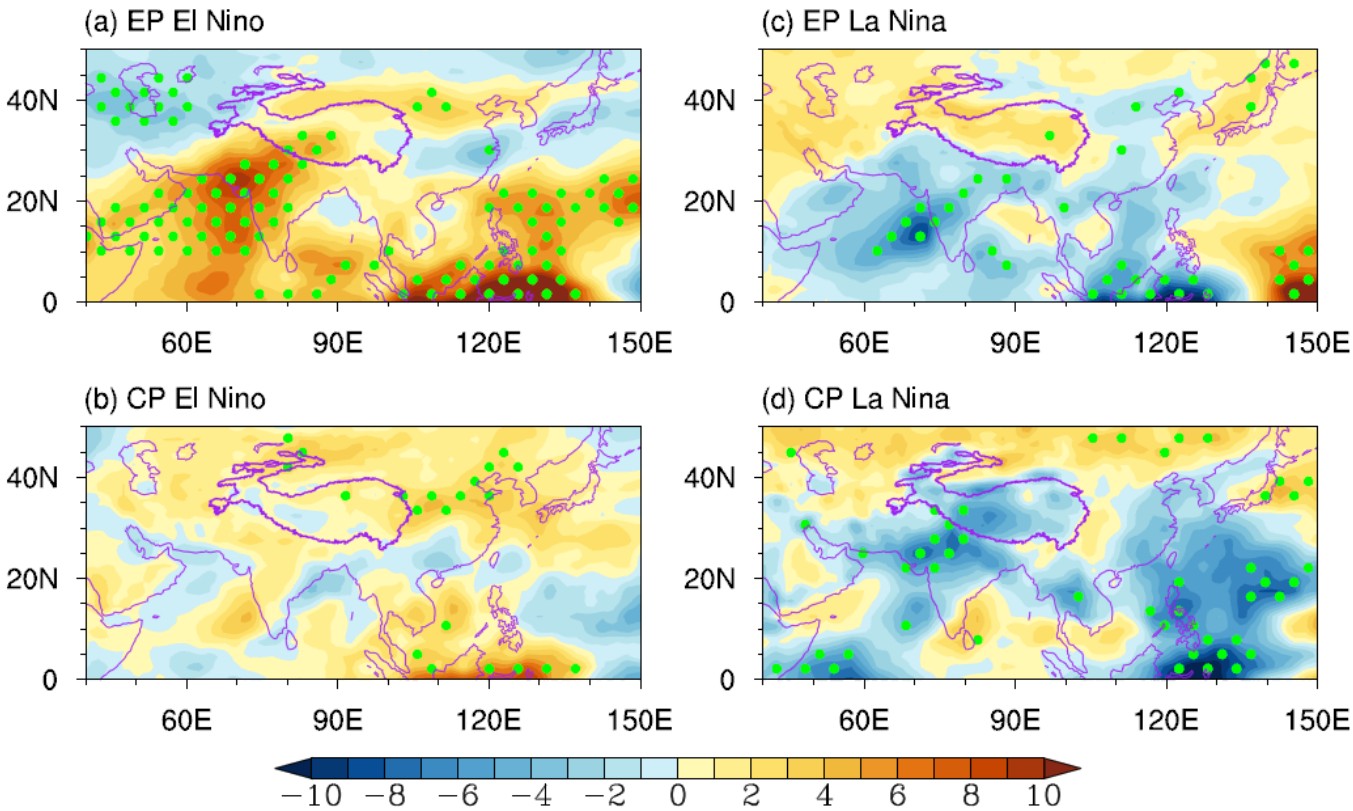

**Figure 6.** Composites of JJAS-mean OLR (shading, unit: W·m$^{-2}$) anomalies for different types of ENSO events. Green dots indicate anomalies statistically significant at the 90% confidence level.

The moisture conditions during the two types of La Niña are not simply symmetric to those observed during El Niños. As shown in Figure 5c, there is no significant anomalous vapor flux over the TP. South of the TP, intense northward vapor flux anomalies occur, but they mostly converge over the Himalayan foothills. Only a tiny amount of anomalous moisture can cross the Himalayas and reach the SWTP. Therefore, the lack of a significant precipitation increase in the SWTP can be attributed to unfavorable moisture conditions, despite the potential presence of abnormal ascending motion. In contrast, during the CP La Niña, subtropical easterly winds associated with equivalent-barotropic cyclonic circulation anomalies lead to strong westward vapor flux over the TP. This flux transports abundant moisture anomalies from eastern China and the western North Pacific to the SWTP (Figure 5d). In addition to the regional anomalous ascending motion, the resulting moisture convergence anomaly also contributes to the enhanced summer precipitation in the SWTP during the CP La Niña.

Based on the above analysis, we have concluded that various vertical motion and moisture conditions are responsible for the distinct TP precipitation anomaly during different types of ENSOs. We further apply an atmospheric moisture budget analysis within the domain region of the precipitation anomaly to examine the specific role of these vertical motion and moisture conditions. As shown in Figure 7a, SWTP summer precipitation reduction associated with the EP El Niño can be primarily attributed to anomalous vertical moisture advection: $- < \omega \frac{\partial q}{\partial p} >'$. Moreover, the climatological moisture advected by descending anomalies, $- < \omega' \frac{\partial \overline{q}}{\partial p} >$, and the anomalous moisture advected by climatological vertical motion, $- < \overline{\omega} \frac{\partial q'}{\partial p} >$, are comparable, suggesting that EP El Niño-related descending motion and moisture deficiency contribute to the precipitation decline in the SWTP to a similar extent. For the positive CETP precipitation anomaly associated with CP El Niño, anomalous vertical moisture advection, $- < \omega \frac{\partial q}{\partial p} >'$, remains the dominant contributor,

but the climatological moisture advected by descending anomalies, $-<\omega'\frac{\partial\bar{q}}{\partial p}>$, singly contributes to the CETP precipitation decrease (Figure 7b). This diagnostic result agrees with the above analysis based on the composites; that is, the descending anomalies induced by the CP El Niño play a dominant role in precipitation reduction, but the effect of moisture transport is negligible (Figures 5b and 6b). Evaporation also makes a positive contribution.

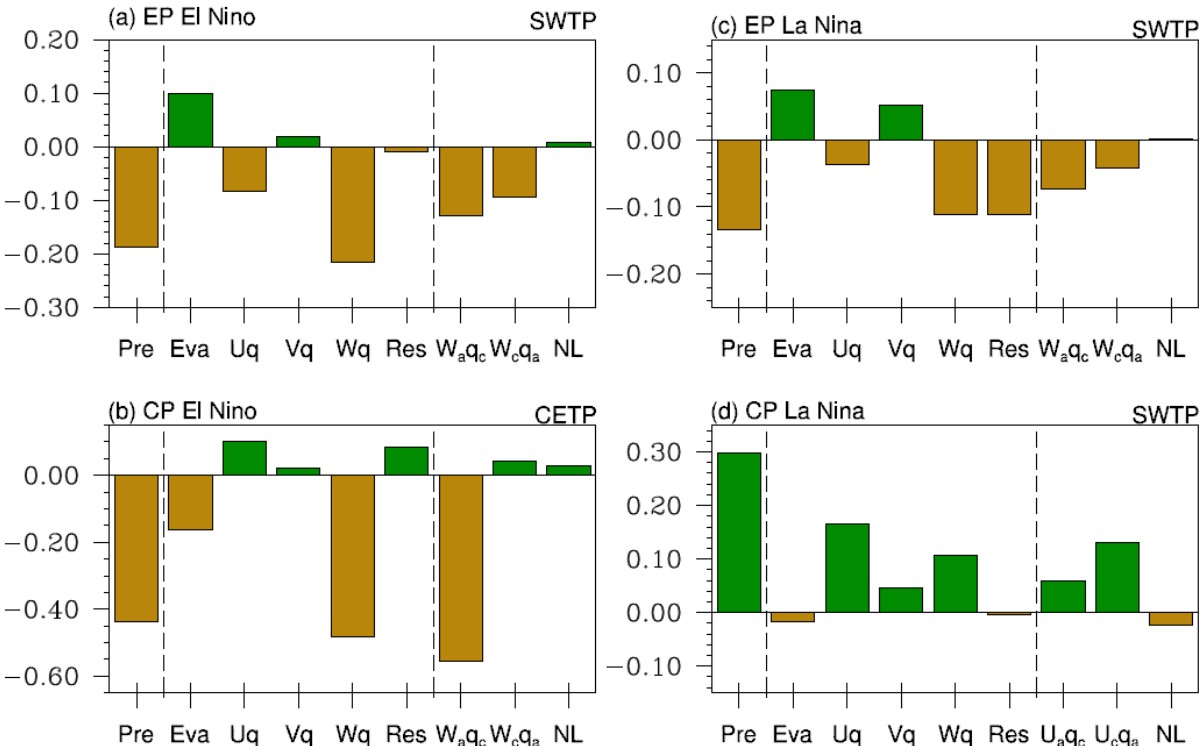

**Figure 7.** Moisture budget analysis of JJAS-mean atmospheric moisture (**a**) for the SWTP region associated with the EP El Niño, (**b**) the CETP region associated with the CP El Niño, (**c**) the SWTP region associated with the EP La Niña, and (**d**) the SWTP region associated with the CP La Niña. The left six terms in turn represent the precipitation anomaly, the evaporation anomaly, the anomalous zonal moisture advection, the anomalous meridional moisture advection, the anomalous vertical moisture advection, and the residual. The three rightmost terms in (**a–c**) are the decomposition of the anomalous vertical moisture advection, namely, the linear term related to the anomalous vertical motion, the linear term related to the moisture variation, and the nonlinear term. Similarly, in (**d**), they represent the linear and nonlinear decomposition terms of anomalous zonal moisture advection. The green and brown colors denote positive and negative anomalies, respectively. Terms associated with the EP (CP) El Niño are obtained by regressing them against the JJAS-mean CTI (WPI) index in all El Niño years, and those for La Niña are similarly obtained and multiplied by a factor of −1. Units are mm·day$^{-1}$·°C$^{-1}$.

The EP La Niña-associated residual term in the moisture budget within the SWTP is almost equivalent to the precipitation variability (Figure 7c), suggesting a less robust relationship between SWTP precipitation and the EP La Niña. This result is reasonable as the precipitation anomaly in the composite is insignificant (Figure 1e). The primary cause of the increased summer precipitation in the SWTP associated with the CP La Niña is the anomalous zonal moisture advection $-<u\frac{\partial q}{\partial x}>'$, and the most significant positive contributor to this advection is the anomalous moisture advected by climatological zonal wind $-<\bar{u}\frac{\partial q'}{\partial x}>$ (Figure 7d). This result confirms the above analysis that emphasizes the role of large-scale moisture transport in enhancing SWTP precipitation during the CP La Niña (Figure 5d). In addition, the anomalous vertical moisture advection, $-<\omega\frac{\partial q}{\partial p}>'$,

also has a positive effect on the precipitation, which mainly results from the anomalous ascending motion.

To summarize, different types of ENSOs trigger various teleconnections during their developing summers, which alter the vertical motion and moisture conditions that affect TP precipitation. Generally, the EP El Niño can induce a circulation pattern resembling the ISM-EASM teleconnection over subtropical Eurasia, which results in significant descending motion as well as moisture deficiency over the SWTP region, together reducing the regional precipitation. Similarly, the CP La Niña can induce an ISM-EASM teleconnection in the opposite phase, which transports a large amount of anomalous moisture to the SWTP region and causes ascending motion, increasing the regional precipitation. In contrast, the CP El Niño mainly stimulates an EAP-like teleconnection pattern over East Asia, which tends to reduce the summer precipitation in the CETP region mainly by leading to a strong regional descending motion anomaly. Finally, the relationship between the EP La Niña and the vertical motion and moisture anomalies over the TP is less robust, although there may also be an ISM-EASM-like teleconnection.

## 4. Discussion

The causes of these ENSO teleconnection variations during different types of ENSOs are complex and multifaceted. As mentioned above, the ISM convection anomaly is critical to triggering the ISM-EASM teleconnection. It is well known that the EP El Niño can induce a Walker circulation anomaly, leading to strong downward motion over the Indo-western Pacific regions (Figure 6a), thus weakening the ISM convection [36]. Meanwhile, the EP El Niño tends to be accompanied by a positive Indian Ocean Dipole (IOD)-like pattern, which contributes to the suppressed ISM as its warming SST anomaly reduces the meridional land-sea thermal contrast (Figure 4a). However, the CP El Niño has less impact on the Indian Ocean and subcontinent (Figures 4b and 6b) because of its weak intensity and westward-shifted warming centre [56,57], which may account for the absence of a clear ISM-EASM teleconnection pattern. Meanwhile, compared with the EP El Niño, the CP El Niño induces a very westward extended western Pacific subtropical high (Figure 4a,b) [47,69], which may lead to anomalous convective activity over the western North Pacific and trigger an EAP-like teleconnection [65].

The EP La Niña also accompanies air-sea anomalies in the Indian Ocean and subcontinent (Figure 4c) because of an enhanced ascending branch of Walker circulation over the Indo-Pacific warm-pool region (Figure 6c). As a result, the enhanced convection associated with the ISM over the Indian subcontinent will excite an ISM-EASM teleconnection featuring equivalent barotropic anticyclonic anomalies in the subtropics. However, as shown in Figure 3c, there is no significant upper-tropospheric negative geopotential anomaly over the northern Indian Ocean during the EP La Niña. This may be because the Pacific cooling anomaly of the EP La Niña is located east, and some of the selected samples are relatively weak. Therefore, it cannot trigger a strong Kelvin wave response in the upper troposphere over the tropical Indian Ocean.

In contrast, the CP La Niña-associated Pacific SST anomaly is located further west and has a much broader meridional scale [47]. Despite having a similar or weaker intensity compared with the EP La Niña, the CP La Niña can induce extensive negative geopotential anomalies over the Indo-Pacific warm-pool region. Meanwhile, although the ascending branch of the CP La Niña-associated anomalous Walker circulation is confined to the Pacific sector (Figure 6d), there are still negative SST anomalies in the northern Indian Ocean, which serve to enhance the ISM and induce an ISM-EASM teleconnection featuring anticyclonic circulation anomalies. We note that all CP La Niña events selected in this study follow another La Niña in the preceding winter. Thus, those negative SST anomalies in the northern Indian Ocean may also be attributed to the preceding La Niñas through the Indo-western Pacific Ocean capacitor mechanism [28,29]. The above discussion provides some possible explanations for the varying teleconnection patterns during different types of ENSOs based on composites of limited historical events. However, these processes

are not mutually exclusive, and sometimes, these teleconnection patterns may coexist. Therefore, further studies are needed to identify the critical factor that determines whether the ISM-EASM teleconnection or the EAP teleconnection dominates during different types of ENSOs.

Previous studies have revealed that the ENSO regime and its connection with the East Asian climate have experienced a decadal shift [70–72]. Whether the teleconnection impacts on TP summer precipitation also exhibit similar variations and what the underlying mechanisms are need to be investigated in the future. This study highlights the importance and necessity of considering spatial diversity in investigating ENSO impacts on the TP climate. The newly found significant impact of the CP El Niño on CETP precipitation provides more sources of predictability for TP summer precipitation, which may have implications for regional climate prediction, especially since the CP El Niño tends to occur frequently under global warming [44]. In fact, this predictability has been suggested by some numerical simulations and climate prediction products [73,74]. However, whether these distinct ENSO teleconnection impacts and underlying mechanisms can be reasonably reproduced in climate models or numerical climate prediction systems remains unclear and deserves further investigation.

## 5. Conclusions

Based on multiple precipitation data and satellite-observed and reanalyzed circulation data, this study investigated the distinct impacts of the two spatial types of ENSOs on TP summer precipitation and analyzed the associated atmospheric teleconnections, reaching the following conclusions.

(1) During their developing summers, both the EP and CP types of ENSOs exhibit asymmetric impacts on TP summer precipitation among El Niños and La Niñas, and the impacts of the EP El Niño and the CP La Niña are trans-type inverse.

(2) SWTP summer precipitation significantly decreases during the EP El Niño but increases during the CP La Niña, while CETP summer precipitation decreases intensely during the CP El Niño. No significant anomaly occurs during the EP La Niña.

(3) The distinct variation in TP summer precipitation during different types of ENSOs can be attributed to various teleconnections that provide altered vertical motion and moisture conditions. The EP El Niño and the CP La Niña both trigger an atmospheric response resembling the ISM-EASM teleconnection but in opposite phases, while the CP El Niño induces an EAP-like teleconnection pattern.

**Supplementary Materials:** The following supporting information can be downloaded at: https://www.mdpi.com/article/10.3390/rs15164030/s1, Figure S1: Tibetan Plateau summer precipitation anomaly during different types of ENSOs across multiple datasets.

**Author Contributions:** Conceptualization, H.-L.R.; methodology, H.-L.R. and M.L.; software, M.L.; validation, M.L.; formal analysis, H.-L.R. and M.L.; investigation, H.-L.R., M.L. and R.W.; writing—original draft preparation, M.L.; writing—review and editing, M.L., H.-L.R., J.M. and X.M.; supervision, H.-L.R.; funding acquisition, H.-L.R., M.L. and R.W. All authors have read and agreed to the published version of the manuscript.

**Funding:** This research was jointly funded by China National Natural Science Foundation (grant numbers U2242206, 42205047, 42105067, and 41975094) and the Special operating expenses of scientific research institutions for "Key Technology Development of Numerical Forecasting" of Chinese Academy of Meteorological Sciences.

**Data Availability Statement:** CRU data are openly available at https://www.uea.ac.uk/web/groups-and-centres/climatic-research-unit/data (accessed on 18 May 2023), and APHRODITE precipitation data are at http://aphrodite.st.hirosaki-u.ac.jp/index.html (accessed on 18 May 2023). GPCC precipitation and NOAA OLR data can be found at https://psl.noaa.gov/data/gridded/index.html (accessed on 21 May 2023). HadISST data can be found at https://www.metoffice.gov.uk/hadobs/index.html (accessed on 20 February 2023). ERA5 data are at https://cds.climate.copernicus.eu/ (accessed on 18 March 2023). CN05.1 precipitation and the station-observed precipitation were

obtained by the authors through an agreement with the China Meteorological Administration for the current study and are not publicly available.

**Acknowledgments:** We thank the National Meteorological Information Center of China Meteorological Administration for providing precipitation data.

**Conflicts of Interest:** The authors declare no conflict of interest.

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
