# Peer review of "Distinct Impacts of Two Types of Developing El Niño–Southern Oscillations on Tibetan Plateau Summer Precipitation"

_remotesensing, doi:10.3390/rs15164030_

Round 1

Reviewer 1 Report

minor revisions will be needed

Author Response

Please see the attachment, as our response includes figures that may be better reviewed in Word. Thank you.

Reviewer 2 Report

This study provides a thorough analysis on impact of different types of ENSO on the TP summer precipitation. As is known, TP and ENSO are both important factors that can exert influence on the East Asian monsoon climate. Thus better understanding the relationship between ENSO and TP precipitation can deepen our understanding of ENSO impacts on East Asian summer precipitation and is important to regional short-term climate prediction in summer. I think the work is worthy of publication. I have the following suggestions and a minor issue.

1. Previous studies reveal that ENSO has a decadal modulation by oceanic signals, e.g., Pacific decadal oscillation (PDO) (Lin et al., 2018). Whether the authors checked if the influence of different types of ENSO on TP precipitation also have a decadal variation? I notice that the ENSO events adopted in this study cover the period from 1960s to 2010s. If there are differences between ENSO events before and after late 1970s? As we know the ENSO and East Asian climate experience a decadal shift (e.g., Zhou et al., 2009; Dong and Xue, 2016). Maybe this is a future work.

2. The results of this work are important to improve the climate prediction in summer. In current climate prediction, climate or earth system models are essential tools. Thus whether the mechanisms revealed in this study can be reasonably reproduced in climate models or in numerical climate prediction systems is another important issue. This deserves further investigation.

Minor issue:

The reference No. 61 is a dataset? The authors should check if ERA5 is cited correctly. I notice other papers cite ERA5 as the paper published in QJRMS in 2020 (Hersbach et al., 2020).

References:

Zhou, T., R. Yu, J. Zhang, H. Drange, C. Cassou, C. Deser, D. L. R. Hodson, E. Sanchez-Gomez , J. Li, N. Keenlyside, X. Xin, Y. Okumura. 2009. Why the Western Pacific Subtropical High has Extended Westward since the Late 1970s. Journal of Climate, 22, 2199-2215

Lin R., F. Zheng, X. Dong. 2018: ENSO Frequency Asymmetry and the Pacific Decadal Oscillation in Observations and 19 CMIP5 Models. Adv. Atmos. Sci., 35(5): 495-506https://doi.org/10.1007/s00376-017-7133-z.

Dong X., F. Xue. 2016: Phase Transition of the Pacific Decadal Oscillation and Decadal Variation of the East Asian Summer Monsoon in the 20th Century. Adv. Atmos. Sci., 33(3): 330-338https://doi.org/10.1007/s00376-015-5130-7

Hersbach, H., Bell, B., Berrisford, P., Hirahara, S., Horányi, A., Muñoz-Sabater, J., Nicolas, J., Peubey, C., Radu, R., Schepers, D., Simmons, A., Soci, C., Abdalla, S., Abellan, X., Balsamo, G., Bechtold, P., Biavati, G., Bidlot, J., Bonavita, M., De Chiara, G., Dahlgren, P., Dee, D., Diamantakis, M., Dragani, R., Flemming, J., Forbes, R., Fuentes, M., Geer, A., Haimberger, L., Healy, S., Hogan, R., Hólm, E., Janisková, M., Keeley, S., Laloyaux, P., Lopez, P., Lupu, C., Radnoti, G., de Rosnay, P., Rozum, I., Vamborg, F., Villaume, S., and Thépaut, J.-N.: The ERA5 global reanalysis, Q. J. Roy. Meteor. Soc., 146, 1999–2049, https://doi.org/10.1002/qj.3803, 2020

Author Response

(The authors gave the same response as above.)

Round 2

Reviewer 1 Report

The revised format has been improved and can be considered for potential publication of the Journal Remote Sensing.